# Metastasis-Directed Stereotactic Body Radiotherapy in Prostate Cancer Patients Treated with Systemic Therapy and Undergoing Oligoprogression: Report on 11 Consecutive Cases

Emanuele Chioccola [1], Mara Caroprese [1], Christina A. Goodyear [1], Angela Barillaro [1], Caterina Oliviero [1], Stefania Clemente [1], Chiara Feoli [1], Luigi Formisano [2], Antonio Farella [1], Laura Cella [3], Manuel Conson [1] and Roberto Pacelli [1,*]

1   Department of Advanced Biomedical Sciences, University Federico II, 80131 Napoli, Italy; emanuele.chioccola@unina.it (E.C.); mara.caroprese@unina.it (M.C.); christina.goodyear3@gmail.com (C.A.G.); angela.barillaro@unina.it (A.B.); caterina.oliviero@unina.it (C.O.); stefania.clemente@unina.it (S.C.); chiara.feoli@unina.it (C.F.); antonio.farella@unina.it (A.F.); manuel.conson@unina.it (M.C.)
2   Department of Clinical Medicine and Surgery, University Federico II, 80131 Napoli, Italy; luigi.formisano1@unina.it
3   Institute of Biostructures and Bioimaging, National Council of Research, 80131 Napoli, Italy; laura.cella@cnr.it
*   Correspondence: roberto.pacelli@unina.it; Tel.: +39-0817462042

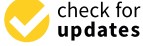



**Simple Summary:** Patients with metastatic prostate cancer undergoing progression in a few metastatic sites (oligoprogression) while undergoing systemic treatment may delay the switch to different drugs and prolong the benefit of the ongoing treatment by metastasis-directed therapy. In this study, we report our experience in treating prostate cancer metastatic oligoprogressive patients with stereotactic body radiotherapy to delay the switch to the next treatment line. This strategy allowed us to prolong the administration of the ongoing systemic therapy by about 1 year.

**Abstract:** Background: Stereotactic body radiotherapy (SBRT) targeted at metastatic sites of disease progression is emerging as a potential therapeutic approach for managing oligoprogressive prostate cancer. However, a definitive benefit has yet to be demonstrated. Herein, we present our institution's experience with this treatment approach. Methods: From April 2018 to March 2023, 11 patients affected by oligoprogressive prostate cancer were treated with SBRT targeting the nodal or bone sites of progression while maintaining the ongoing systemic therapy. Three patients were undergoing single-agent ADT (Androgen Deprivation Therapy), while the remaining eight were receiving a subsequent line of systemic therapy. All patients were evaluated with a pre-treatment 68Ga-PSMA-11 or 18F-fluorocholine PET/CT, which demonstrated between one and five localizations of disease. All the active sites were treated with SBRT in one (15–24 Gy) or three (21–27 Gy) fractions, except for one patient, who was treated in five fractions (35 Gy). PSA serum levels were tested at baseline, one month after RT and at least every three months; all patients underwent a post-treatment 68Ga-PSMA-11 or 18F-fluorocholine PET/CT. The evaluated endpoints were PSA response, defined as a post-treatment decrease >50% from baseline measured within 6 months, time to next-line systemic treatment (NEST), local control (LC), biochemical progression-free survival (bPFS), radiological progression-free survival (rPFS) and freedom from polymetastatic progression (FPP). Results: Nineteen lesions were treated (seven nodal and twelve bone). At a median follow-up of 19 months (7–63), 9 of the 11 patients had a PSA response; all patients had local control of the treated metastases. A total of six patients switched to a next-line systemic treatment, with a median NEST of 13 months. Six patients had polymetastatic progression with an FPP median time of 19 months. No patients died during the follow-up period. The SBRT-related toxicity was negligible. Conclusions: Our data support the use of SBRT targeting the sites of oligoprogressive disease before moving to a subsequent line of systemic treatment in patients with metastatic prostate cancer. Prospective studies to evaluate the potential impact of this approach on overall survival are warranted.

**Keywords:** metastatic prostate cancer; oligoprogression; stereotactic body radiotherapy

## 1. Introduction

Prostate cancer (PCa) is one of the most prevalent malignancies and a leading cause of cancer-related mortality in Western countries, with important implications for global health [1,2]. In this scenario, metastatic castration-resistant prostate cancer (mCRPC) represents a significant clinical challenge, accounting for an estimated 258,400 global deaths annually [3,4].

Despite advances in therapeutic strategies, where systemic treatment serves as the cornerstone of management, mCRPC is characterized by clinical progression despite testosterone levels below 50 ng per deciliter (1.7 nmol per liter) and a predictable sequence of events leading to death within 24 to 48 months of castration resistance onset [5,6].

In recent years, the evolving landscape of metastatic prostate cancer has revealed a distinct clinical subset—oligometastatic prostate cancer. This intermediate state, first conceptualized by Hellman and Weichselbaum in 1995, represents a critical phase between localized and widespread metastases, characterized by a limited number of secondary lesions [7–9]. In the evaluation and staging of advanced prostate cancer imaging, an important role is currently played by the PSMA-PET (with 68Ga or 18F) enabling highly sensitive detection of metastatic sites and a reliable assessment of the disease burden.

The definition of oligometastatic disease remains a subject of debate, going from three to five bone or nodal metastases while excluding visceral localizations. In recent years, another classification of metastatic prostate cancer has been suggested by the CHAARTED trial [10], distinguishing between low- and high-volume disease, with the latter being defined by the presence of visceral metastases or of more than four bone lesions, including at least one outside the vertebral column or pelvis.

Oligometastatic prostate cancer is further stratified into three subgroups: *de novo* (synchronous), oligorecurrent (metachronous) and oligoprogressive (one or few sites of disease progression during systemic therapy), each shedding light on the dynamic nature of the disease [11–13]. These subgroups offer valuable insights into the disease progression and response to treatment. While de novo and oligorecurrent mPC could still be defined as castration-sensitive diseases, oligoprogressive mPC has unavoidably developed a castration resistance over time and is associated with a decline in prognosis and diminished systemic treatment options. In this group of patients, the therapeutic options are limited, ranging from chemotherapy with Docetaxel or Cabazitaxel to Abiraterone acetate or androgen receptor-targeted agents (ARTA) such as enzalutamide and apalutamide [14–17].

Despite ongoing investigations into the definitive benefits of managing oligometastatic disease, the application of stereotactic body radiotherapy (SBRT) or other localized therapies for all active lesions has emerged as a promising strategy [18–21].

The SABR-COMET study [21] was the first randomized trial demonstrating the impact of ablative therapy on a primary end point of OS in patients with oligometastasis. A randomized phase II screening design was employed to investigate the impact of stereotactic ablative radiotherapy on OS in patients with a controlled primary malignancy, from different sites and one to five metastatic lesions. Patients were randomized in a 1:2 ratio between standard-of-care treatments alone and standard-of-care plus SABR. The 5-year OS rate demonstrated a significant difference, with 17.7% in arm 1 and 42.3% in arm 2. Importantly, the extended follow-up revealed a larger impact of SABR on OS than in the initial analysis, and this effect was durable over time.

Particularly in the context of oligoprogressive CRPC, the rationale for employing SBRT lies in its potential to enhance disease control and influence metastatic behavior, presenting a compelling avenue for exploration and therapeutic intervention [22–25]. For patients affected by mCRPC, few active therapeutic lines are available, and the time of disease control that is achieved by a single treatment line contributes to the overall survival time.

The strategy of using metastasis-directed local therapy to extend the duration of disease control could represent a valuable tool to enhance the effectiveness of systemic therapy for oligoprogressive mCRPC patients.

In this paper, we explore the potential role of SBRT as a promising adjunct to systemic therapies. By reporting on a small group of oligoprogressive metastatic prostate cancer patients who were consecutively treated at our department, our aim is to contribute to improvement in the management of this challenging subset of prostate cancer.

## 2. Patients and Methods

We retrospectively evaluated patients who were affected by oligoprogressive PC and who underwent metastasis-directed SBRT, while maintaining the ongoing systemic therapy, at AOU "Federico II" of Naples, Italy. Eligible criteria included age $\geq$ 18 years; pathologically confirmed PC; biochemical and radiological relapse of disease during systemic treatment; from one to five bone or nodal metastases, diagnosed with a 68Ga-PSMA or 18F-Choline PET-CT; having been discussed by a multidisciplinary board including an urologist, a medical oncologist and a radiation oncologist; and a follow-up time after RT of at least six months. All the active sites of disease must have been treated with ablative SBRT. Patients were excluded if they had switched to a next-line systemic therapy in the previous three months, in case of high-volume metastatic disease according to the CHAARTED criteria, and if they had been treated on a lesion that had already undergone RT.

Each patient underwent a simulation CT with 3 mm thick slices; immobilization devices varied based on the target location and patient's clinical conditions. CT images were transferred to MIM Maestro® contouring software version 6.6.7 and then to Pinnacle PHILIPS TPS software version 9.10. The gross tumor volume (GTV) was delineated, combining morphological and metabolic information, fusing the simulation CT and the pre-treatment PET/CT images (see Figure 1). No additional margin was added for microscopic spread of disease, and the GTVs were expanded by a maximum of 5 mm to define the corresponding planning target volumes (PTVs), accounting for organ motion and setup errors.

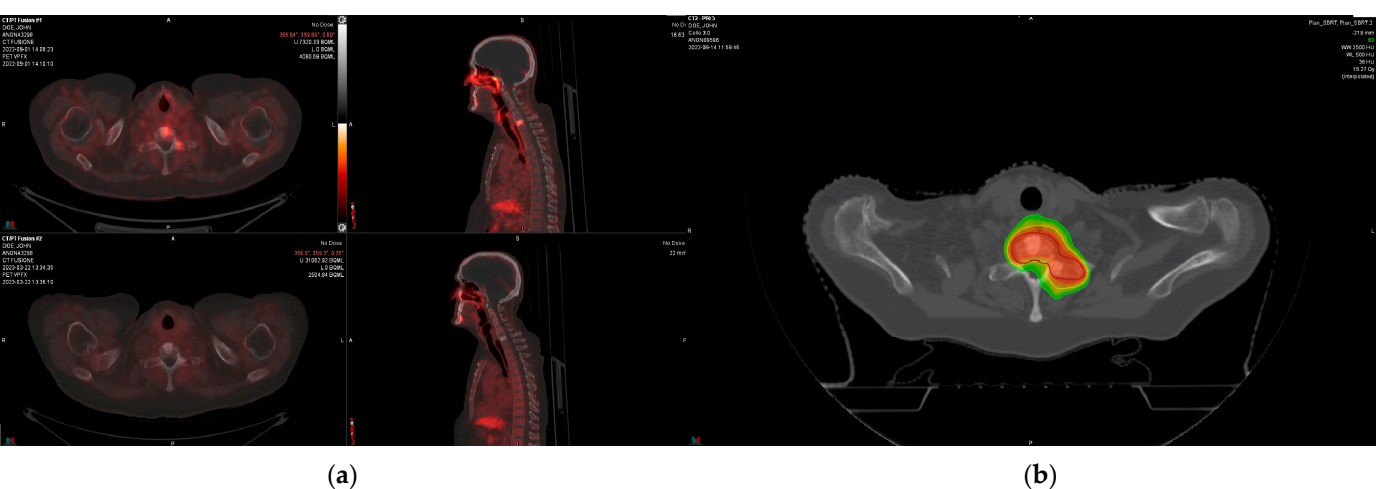

(**a**)  (**b**)

**Figure 1.** (**a**) Pre-treatment (left) and post-treatment (right) PET/CT of a vertebral lesion; (**b**) axial image of the SBRT treatment dose distribution.

All the active sites of disease were treated with SBRT, and dose prescriptions were based on the size and location of the target lesions. The treatments were delivered by Varian TrueBeam STx version 2.0.

RT was delivered daily from Monday to Friday, and each treatment fraction was preceded by a cone-beam CT scan for target verification, evaluated by a department radiation oncologist.

PSA serum levels were tested at baseline, one month after RT and at least every three months for all of the follow-up period, combined with contextual clinical examination. All patients underwent a pre- and post-treatment 68Ga-PSMA or 18F-Choline PET/CT; other imaging investigations were prescribed based on clinical indications.

The primary endpoints were PSA response, defined as a decrease >50% from baseline of the serum PSA level measured within 6 months from the end of the MDT, and next-line systemic treatment-free survival (NEST-FS), calculated from the last day of SBRT to the first day of NEST or last follow-up or death. Next-line systemic treatment was administered based on individual evaluation by the referring medical oncologist in case of progression of disease after the metastasis-directed therapy (MDT).

Secondary endpoints were local control (LC) of treated metastases, defined as the time from the beginning of therapy to the progression of the treated metastases; biochemical progression-free survival (bPFS), meaning the interval between the first fraction of SBRT and the PSA progression, defined according to PCWG2 recommendations as a 25% increase from the baseline value, along with an increase in absolute value of 2 ng/mL or more; radiological progression-free survival (rPFS), defined as the interval between the first fraction of SBRT and the progression of disease on a PSMA-PET evaluation, according to the PERCIST 1.0 criteria, being a $\geq$30% increase in highest SUV max of the baseline metastases and/or the detection of new PSMA avid lesions; freedom from polymetastatic progression (FPP), defined as the time from the first administration of radiotherapy to polymetastatic progression, meaning the development of high-volume disease according to the CHAARTED criteria. For all these endpoints, if the event did not occur, censoring was the date of the last follow-up or death.

## 3. Results

Over the observational period spanning from April 2018 to March 2023, our study enrolled a cohort of 11 patients. The median patient age at time of SBRT was 71 years (range 48–82 years). The median PSA level was 0.8 ng/mL (0.17–14.3 ng/mL). This cohort collectively represented a spectrum of 19 treated metastases, including 7 (37%) nodal and 12 (63%) bone metastases, allowing for MDT outcome analysis. Patients were followed up for a median time of 19 months (range 7–63 months).

Three patients presented initially with metastatic disease but subsequently exhibited a transition to oligoprogression during systemic treatment.

At the time of the start of MDT, patients displayed diverse treatment histories, with four patients undergoing third-line, three on second-line, and four on first-line systemic treatments for metastatic disease. The subset of three patients undergoing second-line systemic treatments encountered oligoprogression during single-agent Androgen Deprivation Therapy (ADT) (Table 1).

**Table 1.** Patients and treatment features. ADT (Androgen Deprivation Therapy), Abi (Abiraterone), Enza (Enzalutamide), Apa (Apalutamide), Doce (Docetaxel).

| - | Age | PSA Baseline | Systemic Therapy Line | Ongoing Systemic Therapy | Nodal GTV [N] | Bone GTV [N] |
|---|---|---|---|---|---|---|
| **pt1** | 71 | 8.2 | II | ADT + Abi | 0 | 1 |
| **pt2** | 74 | 14.3 | III | ADT + Enza | 2 | 3 |
| **pt3** | 68 | 0.6 | III | ADT + Doce | 1 | 1 |
| **pt4** | 65 | 0.5 | II | ADT + Abi | 0 | 1 |
| **pt5** | 48 | 0.5 | III | ADT + Enza | 0 | 1 |
| **pt6** | 74 | 0.8 | I | ADT | 1 | 1 |
| **pt7** | 64 | 0.6 | I | ADT | 1 | 0 |
| **pt8** | 82 | 1.3 | II | ADT + Abi | 0 | 2 |
| **pt9** | 73 | 14 | III | ADT + Abi | 0 | 2 |
| **pt10** | 76 | 1.1 | I | ADT | 1 | 0 |
| **pt11** | 58 | 0.2 | I | ADT + Apa | 1 | 0 |

The pre-treatment 68Ga-PSMA-11 PET/CT imaging was used in six patients and 18F-fluorocholine PET/CT in five; throughout the follow-up duration, 68Ga-PSMA-11 PET/CT was employed in eight cases and 18F-fluorocholine PET/CT in the remaining instances.

The median SBRT total dose was 25.5 Gy (range 15–35 Gy), delivered in one (total dose of 15–24 Gy) or three (21–27 Gy) fractions, except for one patient, who was treated in five fractions (35 Gy). The median biological effective dose ($BED_3$) was 108 Gy.

In Tables 2 and 3, the oncological outcomes that were observed in the cohort of patients are reported. A PSA response was robustly observed in nine patients, reflecting the effectiveness of the therapeutic intervention. However, six individuals eventually transitioned to next-line systemic treatment due to biochemical and/or radiological disease progression, with a calculated median NEST-FS time of 13 months. In particular, biochemical and radiological progression manifested in six and seven cases, respectively, with corresponding median times for bPFS and rPFS recorded at 13 and 12 months.

**Table 2.** Outcome events. PSA resp (PSA response), B-rel (biochemical relapse), R-prog (radiological progression), P-met prog (polymetastatic progression), LC (local control), NEST (next-line systemic treatment).

| PATIENT | PSA Resp | B-Rel | R-Prog | P-Met Prog | LC | NEST |
|---------|----------|-------|--------|------------|-----|------|
| pt1 | Yes | Yes | Yes | Yes | Yes | Yes |
| pt2 | Yes | Yes | Yes | Yes | Yes | Yes |
| pt3 | No | Yes | Yes | No | Yes | Yes |
| pt4 | Yes | Yes | Yes | Yes | Yes | Yes |
| pt5 | No | Yes | Yes | Yes | Yes | Yes |
| pt6 | Yes | No | No | No | Yes | No |
| pt7 | Yes | No | No | No | Yes | No |
| pt8 | Yes | No | No | No | Yes | No |
| pt9 | Yes | Yes | Yes | Yes | Yes | Yes |
| pt10 | Yes | No | No | No | Yes | No |
| pt11 | Yes | No | Yes | Yes | Yes | No |
| N. | 9 | 6 | 7 | 6 | 11 | 6 |

**Table 3.** Outcome events' timing. FUP (follow up), b-PFS (biochemical progression-free survival), r-PFS (radiological PFS), FPP (freedom from polymetastatic progression), LC (local control), NEST-FS (next-line systemic treatment-free survival).

| - | FUP | b-PFS | r-PFS | FPP | LC | NEST-FS |
|---|-----|-------|-------|-----|-----|---------|
| pt1 | 63 | 23 | 25 | 61 | 63 | 35 |
| pt2 | 25 | 9 | 6 | 24 | 25 | 10 |
| pt3 | 9 | 4 | 4 | 9 | 9 | 6 |
| pt4 | 32 | 18 | 19 | 19 | 32 | 19 |
| pt5 | 14 | 6 | 10 | 10 | 14 | 7 |
| pt6 | 19 | 19 | 19 | 19 | 19 | 32 |
| pt7 | 50 | 50 | 50 | 50 | 50 | 14 |
| pt8 | 12 | 12 | 12 | 12 | 12 | 19 |
| pt9 | 13 | 13 | 12 | 12 | 13 | 13 |
| pt10 | 61 | 61 | 61 | 61 | 61 | 12 |
| pt11 | 7 | 7 | 3 | 3 | 7 | 7 |
| Median time | 19 | 13 | 12 | 19 | 19 | 13 |

The development of high-volume metastatic prostate cancer after SBRT occurred in six patients, with a median FPP time of 19 months. Importantly, all patients demonstrated satisfactory local control over the treated metastases.

## 4. Discussion

The data reported in the present study suggest a possible benefit of treating the few sites of disease progression with SBRT while maintaining the ongoing systemic therapy

in mCRPC patients. A PSA response was obtained in most patients, suggesting a "true" oligoprogression. In addition, optimal local control of all the treated metastases over time was achieved. Of note, in almost half of the patients, biochemical control was maintained during the entire follow-up time. Combined with a similar proportion of radiological control, approximately one in two patients avoided switching to a subsequent therapeutic line and could prolong the use of the ongoing therapy. With the limits of the number of patients and the retrospective nature of the study, this emerges as the primary advantage of this approach. Particularly noteworthy is the fact that, in our series, most of the patients were undergoing a second- or third-line systemic treatment. Of note, all three patients in the first line of therapy with only ADT are at the present free of progression. Despite toxicity not being an endpoint of the study, no relevant adverse effects according to the Radiation Therapy Oncology Group (RTOG)'s acute and late toxicity grading have been registered. This finding is consistent with data in the literature demonstrating the feasibility of the SBRT and its favorable impact on the patient's quality of life [23,24,26]

The standard of care for metastatic prostate cancer patients remains systemic therapy, with a local approach to the metastases reserved for palliative intent; this paradigm is applied both in de novo mPC and in the case of progression during systemic treatment. In recent years, MDT with ablative intent as an approach to oligo-metastatic disease is emerging in different type of cancers, and its role in clinical practice is consolidating.

For oligo-metastatic PC, only two randomized controlled trials (RCT's) have been published [23,25,27], both phase II trials analyzing castration-sensitive PC treated with SABR compared with surveillance, without any systemic treatment allowed. Oligo-metastatic status was defined as a maximum of three sites of disease, and both trials showed an advantage in the experimental arms. In the study by Ost et al. [23], with 31 patients in each of the two arms [28], an ADT-free survival of 21 months vs. 13 months was reported in favor of the RT group, without >G1 toxicities and a negative impact on the quality of life; of note, patients with only one metastatic site of disease were double in number in the SABR arm compared to surveillance arm. The ORIOLE trial [25] randomized 36 patients to SABR vs. 18 to observation, reporting an advantage in progression-free survival and an optimal local control in the experimental arm.

Another prospective, non-randomized, clinical trial that assessed the benefit of MDT in oligo-metastatic PC was published in 2021 by Hölscher et al. [24], with a total of 63 patients without ongoing systemic therapy: 70% were treated with SBRT, 22% with conventional fractionated RT (50 Gy in 25 fx) and 8% with both of them. Half of the patients did not start ADT after two years of RT; only 24.6% did not reach a PSA response after RT; and biochemical progression happened in 47 patients, with a median time of 13.2 months.

In the castration-resistant oligo-progressive setting, there are no prospective clinical trials evaluating the role of MDT in addition to the systemic therapy [29,30]. Several retrospective studies are available in the literature [26,28,31–39].

In 2019, Berghen et al. reported their experience with 30 patients, 3 of which were non-metastatic loco-regional recurrence and had undergone MDT with surgery or radiotherapy while maintaining the ongoing systemic therapy [34]. With a follow-up time of 18 months, the total PFS- and NEST-free survival median times were 10 and 16 months, respectively; excluding the 3 loco-regional recurrence cases and the ones treated with non-ablative RT, 20 patients treated with metastasis surgical resection or SBRT had a NEST-FS median time of 21 months.

In a study by Yoshida et al. [37], the researchers aimed to assess the efficacy of progressive site-directed therapy for oligoprogressive castration-resistant prostate cancer. The study included a cohort of 101 CRPC patients who underwent whole-body diffusion-weighted magnetic resonance imaging, coinciding with the consideration of a new line of therapy. Radiation therapy alongside the continuation of unchanged systemic therapy was recommended. The study demonstrated that progressive site-directed therapy, particularly when localized to intrapelvic lesions, resulted in a significant PSA response, with

a 50% decline observed in 70% of cases, while the median time to PSA progression was 8.7 months.

An Italian retrospective multicenter study was conducted by Triggiani et al. [38], including 86 patients and 117 treated metastases from eleven centers. All the evaluated individuals were affected by oligoprogressive PC, with one to five bone or nodal sites of disease, all treated with SBRT; of note is that imaging evaluation was performed with PET/CT or with CT and bone scan. Enrolled patients were in disease progression during ADT. The use of Abiraterone, ARTA or chemotherapy was considered an exclusion criterion. With a median follow-up time of 30.7 months, local control was 80%, while the median new metastasis-free survival after SBRT was 12.3 months, and the one- and two-year distant progression-free survival was 52.3% and 33.7%, respectively; the median systemic treatment-free survival was 21.8 months, and the one-year systemic treatment-free survival was 72.1%.

In 2020, a retrospective analysis of the Johns Hopkins Hospital and of the Mayo Clinic databases were published, reporting data from 68 mCRPC patients treated with SABR on all the sites of disease, detected in 70% of cases with a Choline or PSMA PET/CT, maintaining the systemic therapy [35]. With a follow-up median time of 30.9 months, the time to PSA failure and to the detection of new metastases were 9.6 and 10.8 months, respectively, while the time to the start of a new treatment was 15.6 months. Local control was 98% and 86% after 1 and 2 years from the end of RT, respectively.

Ingrosso et al. reported data on 34 patients in progression during ADT plus ARTT and treated with SBRT on the metastatic lesions, mostly detected with Choline-PET/CT [26]: at a median follow up time of 25 months, 13 patients did not switch systemic therapy, and the total rPFS and NEST-FS times were 13.4 and 16.9 months respectively.

A retrospective study conducted by Franzese et al. and published in 2022 [28] evaluated 53 patients who were affected by oligoprogressive castration-resistant PC and had undergone metastases-directed SBRT. The enrolled patients were generally unfit for the intensification/switch of systemic therapy or were affected by indolent disease, characterized by slow PSA kinetics and a long progression-free interval. Approximately 79% of them were assuming antiandrogen–LHRH agonist/antagonist systemic therapy, while 21% were receiving ongoing ARTA treatment; 92% of the patients underwent SBRT directed to one or two metastases, with a median BED3 Gy of 117.3 (range 66.6–240). With a median follow-up time of 24.9 months, 1- and 2-year FPP rates of 80.1% and 68.9% were observed, respectively, with prolonged disease-free survival and nodal disease found to be significantly associated with an improved FPP or, conversely, to the ongoing ARTA therapy during SBRT, which was associated with lower FPP rates. The median distant progression-free survival (DPFS) and NEST-FS were 8.9 and 15.2 months, respectively; the 2-year LC and OS rates were 92% and 85.2%.

In 2023, a French multicenter retrospective analysis by Baron et al. [39] evaluated the oncological outcomes after SBRT for oligoprogressive CRPC and investigated prognostic factors for systemic therapy escalation-free survival (STE-FS). A total of 50 patients were enrolled, and most of them (78%) received ablative radiotherapy on a single metastatic site. The median follow-up time was 23 months. The local control and PSA response rates were 88% and 32% (38% of missing data for the post-SBRT PSA assessment), respectively. The median PFS was 13 months, with 68% of the patients experiencing progression. STE-FS, which was the primary endpoint, had a median time of 13.1 months, with 38% of the patients exhibiting stable or decreased post-SBRT PSA; the only prognostic factor that significantly related to STE-FS on a univariate and multivariate analysis was the post-SBRT biochemical response, with an advantage in both the PSA-stable and PSA-responsive cases.

Our study shows some inherent limitations, given its retrospective design, a characteristic that is shared with other reported trials. The absence of randomized controlled prospective trials in this specific setting underscores the potential utility of retrospective studies in offering insights that aid physicians in navigating treatment decisions.

An important constraint lies in the modest number of patients who were included in our evaluation. The low number, while a limitation, also accentuates the scarcity of data in this niche, highlighting the need for expanded research activities in the domain of metastatic oligoprogressive prostate cancer.

Additionally, the follow-up duration in our study, although sufficient for analyzing short- and mid-term effects of the SBRT approach, falls short of affording a reasonable evaluation of patients' overall survival. More protracted observation periods are necessary to draw meaningful conclusions about the impact on survival outcomes. Despite these acknowledged limitations, the study contributes valuable insights within the context of the existing literature, offering a foundation for future prospective investigations and aiding clinicians in navigating the intricate landscape of metastatic oligoprogressive prostate cancer treatment choices.

## 5. Conclusions

In this single-center retrospective analysis, we examined patients with metastatic prostate cancer experiencing oligoprogression during systemic treatment who underwent metastasis-directed radiation therapy. Throughout the treatment, they continued their ongoing systemic therapy and transitioned to a subsequent line only in cases of disease progression. Almost all patients exhibited a positive PSA response, and the entire cohort achieved local control. Notably, approximately half of the patients maintained sustained disease control without altering their systemic treatment during the follow-up period.

While our study's limited sample size is acknowledged, our findings align with the existing literature, suggesting the potential to postpone the initiation of a subsequent line of therapy by roughly one year. However, a more extended follow-up time is necessary to assess the impact on overall survival. The need for prospective randomized clinical trials focusing on metastasis-directed therapy in metastatic castration-resistant prostate cancer remains crucial to thoroughly evaluate the potential survival benefits of this promising therapeutic approach.

**Author Contributions:** Conceptualization E.C., R.P., L.C. and M.C. (Manuel Conson); methodology, E.C., L.F. and M.C. (Mara Caroprese); formal analysis, A.B., M.C. (Mara Caroprese), M.C. (Manuel Conson), L.C., R.P., C.F., S.C. and C.O.; investigation, E.C., C.A.G., A.F., M.C. (Manuel Conson), L.F. and R.P.; data curation, E.C., M.C. (Mara Caroprese), A.B., C.A.G., C.O., S.C., A.F., M.C. (Manuel Conson), C.F. and R.P.; writing—original draft preparation, E.C., C.A.G., L.C., M.C. (Manuel Conson) and R.P.; writing—review and editing, E.C. and R.P.; supervision, A.F., M.C. (Manuel Conson) and R.P. All authors have read and agreed to the published version of the manuscript.

**Funding:** This research received no external funding.

**Institutional Review Board Statement:** Comitato Etico per le Attività Biomediche "Carlo Romano" dell'Università degli Studi di Napoli Federico II; protocol code institutional review board 222-10, 2017.

**Informed Consent Statement:** Informed consent was obtained from all subjects involved in the study.

**Data Availability Statement:** The data presented in this study are available on request from the corresponding author.

**Conflicts of Interest:** The authors declare no conflicts of interest.

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
