# Peer review of "Metastasis-Directed Stereotactic Body Radiotherapy in Prostate Cancer Patients Treated with Systemic Therapy and Undergoing Oligoprogression: Report on 11 Consecutive Cases"

_radiation, doi:10.3390/radiation4020009_

Round 1

Reviewer 1 Report

Comments and Suggestions for Authors

The paper by Pacelli et al, explores the usefulness of SBRT in oligometastatic prostate cancer. The sample size is small, but the included patients reflect the majority of those seen in clinics. The results underline the value of adding SBRT in different clinical scenarios. Study limitations are also declared and clearly settled in discussion and conclusion.

Author Response

We thank the reviewer for the positive comments to our work

Reviewer 2 Report

Comments and Suggestions for Authors

This was a well-presented study on external radiation therapy of patients with oligometastatic prostate cancer. I have a couple of minor issues:

The PET tracers used to determine radiological progression-free survival. For one they are insufficiently described, I assume 68Ga-PSMA is actually 68Ga-PSMA-11 but that should be written out. Also 18F-choline is usually written as 18F-Fluorocholine. Another major point regarding this is that these are different tracers with different targets and performance and can not just be substituted for one another. There should at a minimum be reported how many scans were done with either, and if for example more new metastases were found in patients receiving 68Ga-PSMA-11 scans than with 18F-Fluorocholine?

Another question I would have liked to see in the discussion was if there were any characteristics of the half of patients who maintained a sustained disease control? Number of metastases, location of metastases etc?

Author Response

We thank the reviewer for the comments and for suggestions that improve our paper.

We modified the paper accordingly. You will see in blue the related modifications in the paper.

This was a well-presented study on external radiation therapy of patients with oligometastatic prostate cancer. I have a couple of minor issues:

The PET tracers used to determine radiological progression-free survival. For one they are insufficiently described, I assume 68Ga-PSMA is actually 68Ga-PSMA-11 but that should be written out. Also 18F-choline is usually written as 18F-Fluorocholine.

We modified the definition of the tracers as requested

Another major point regarding this is that these are different tracers with different targets and performance and can not just be substituted for one another. There should at a minimum be reported how many scans were done with either, and if for example more new metastases were found in patients receiving 68Ga-PSMA-11 scans than with 18F-Fluorocholine?

Accordingly with the reviewer request we specified in the “Results” section the number and type of PET/CT done before and after the treatment (page 4, line 169-171). About the number of metastases the small sample of patients limited this kind of analysis. However, even without statistics, no qualitative differences were noticed in our group of patients.

Another question I would have liked to see in the discussion was if there were any characteristics of the half of patients who maintained a sustained disease control? Number of metastases, location of metastases etc?

We specified the features of the patients that maintained the progression free status (page 6, line 203-204). The only apparent observation is that all patients in first line of treatment with ADT only (3 patients) are free of progression.

Reviewer 3 Report

Comments and Suggestions for Authors

The authors presented a paper about "Metastasis Directed Stereotactic Body Radiotherapy In Prostate Cancer Patients Treated With Systemic Therapy And Undergoing To Oligoprogression: Report On 11 Consecutive Cases".

The topic is absolutely intresting both from a clinical and from a reseacrh point of view.

I have a few suggestions to improve the paper:

1) the authors correctly refer to Hellman and Weichselbaum who in 1995 used the definition of olimetastases for the first time however it would useful to cite the further defition such as oligorecurrence proposed by Niibe in 2010 (Jap J Clin Oncol) and then to cite the definition of  oligoprogression 

2) Please state if all patients were discussed within the frame of a multidisciplianry tumor board and figures involved

3) Please clarify the toxicity scale used to collect the data

Author Response

We thank the reviewer for the comments and for suggestions that improve our paper.

We modified the paper accordingly. You will see in green the related modifications in the paper.

The authors presented a paper about "Metastasis Directed Stereotactic Body Radiotherapy In Prostate Cancer Patients Treated With Systemic Therapy And Undergoing To Oligoprogression: Report On 11 Consecutive Cases".

The topic is absolutely intresting both from a clinical and from a reseacrh point of view.

I have a few suggestions to improve the paper:

  • the authors correctly refer to Hellman and Weichselbaum who in 1995 used the definition of olimetastases for the first time however it would useful to cite the further defition such as oligorecurrence proposed by Niibe in 2010 (Jap J Clin Oncol) and then to cite the definition of oligoprogression 

We defined oligoprogression (page 2, line 68-69) and added the suggested citation.

  • Please state if all patients were discussed within the frame of a multidisciplianry tumor board and figures involved

The patients were discussed in a multidisciplinary tumor board including an urologist, a medical oncologist and a radiation oncologist. Accordingly we stated that at page 3, line 107-108

  • Please clarify the toxicity scale used to collect the data

We evaluated the toxicity of radiation treatment with RTOG scale. Accordingly we stated that at page 6, line 207-208